# A Sensorised Glove to Detect Scratching for Patients with Atopic Dermatitis

**DOI:** 10.3390/s23249782

**Published:** 2023-12-12

**Authors:** Cheuk-Yan Au, Syen Yee Leow, Chunxiao Yi, Darrion Ang, Joo Chuan Yeo, Mark Jean Aan Koh, Ali Asgar Saleem Bhagat

**Affiliations:** 1Institute for Health Innovation & Technology (iHealthtech), National University of Singapore (NUS) MD6, 14 Medical Drive, #14-01, Singapore 117599, Singapore; cy.au@nus.edu.sg (C.-Y.A.); y.chunxiao@u.nus.edu (C.Y.); jcyeo@nus.edu.sg (J.C.Y.); 2Department of Dermatology, KK Women’s and Children’s Hospital, 100 Bukit Timah Road, Singapore 229899, Singaporemark.koh.j.a@singhealth.com.sg (M.J.A.K.); 3Department of Biomedical Engineering, National University of Singapore (NUS), 4 Engineering Drive 3, Singapore 117583, Singapore

**Keywords:** atopic dermatitis, dermatology, scratching, wearable sensors, medical device, neural network

## Abstract

In this work, a lightweight compliant glove that detects scratching using data from microtubular stretchable sensors on each finger and an inertial measurement unit (IMU) on the palm through a machine learning model is presented: the SensorIsed Glove for Monitoring Atopic Dermatitis (SIGMA). SIGMA provides the user and clinicians with a quantifiable way of assaying scratch as a proxy to itch. With the quantitative information detailing scratching frequency and duration, the clinicians would be able to better classify the severity of itch and scratching caused by atopic dermatitis (AD) more objectively to optimise treatment for the patients, as opposed to the current subjective methods of assessments that are currently in use in hospitals and research settings. The validation data demonstrated an accuracy of 83% of the scratch prediction algorithm, while a separate 30 min validation trial had an accuracy of 99% in a controlled environment. In a pilot study with children (*n* = 6), SIGMA accurately detected 94.4% of scratching when the glove was donned. We believe that this simple device will empower dermatologists to more effectively measure and quantify itching and scratching in AD, and guide personalised treatment decisions.

## 1. Introduction

Atopic dermatitis (AD), also known as eczema, is a recurrent, itchy skin condition that commonly affects children [1]. Around 15–20% of children and 6–10% of adults are affected by AD worldwide [2]. It is a chronic condition that is associated with family genetics, although it can improve or even clear completely when a child gets older [3]. AD symptoms include itching, redness, small bumps, and dry flaky and scaly skin on small patches or over a large area of the body. People with AD have periods where their symptoms are more severe—a flare-up—and other times when it is less noticeable [4]. A flare-up can be caused by environmental factors, such as contact with allergens, or it could be triggered internally [5]. It is defined as an “Acute, clinically significant worsening of signs and symptoms of AD requiring therapeutic intervention” by the European Task Force of Atopic Dermatitis (ETFAD) [6]. The most common methods used currently for assessing the severity of AD are Severity Scoring of Atopic Dermatitis (SCORAD) [7], the Eczema Area and Severity Index (EASI) [8], and the Peak Pruritus Numerical Rating Scale (PP-NRS) [9]. These methods are similar and qualitative by nature, classifying the patient into different categories of AD severity by observing clinical signs present through the review of patient history or examining the patients. SCORAD requires every patient to be assessed by observing the intensity and extent of the dryness on different parts of their body and a subjective rating of the symptoms that they are experiencing and how it affects their sleep [7]. EASI requires examining the patient’s body visually and assigning scores based on the clinician’s examinations in order to determine the severity of their AD [8]. PP-NRS requires patients to rate the worst itch they have experienced over the past 24 h [9]. All three described methods can be very subjective in scoring with different patients and clinicians [10]. The clinician or patient may overrate or underrate the severity in different sections of SCORAD, EASI, or PP-NRS due to subjective elements or systemic errors such as recall bias causing results to skew either way. This may lead to suboptimal treatment, as clinicians rely on these scores for treatment [11].

Recently, scratching is monitored as a way to evaluate the degree of itch, and itch is an indication the severity of skin diseases like AD [12,13]. Devices such as the ADAM sensor [14] or the MetaMotionR-based fabric wristband [15] are aiming to quantify scratching as an assessment of the severity of skin diseases. Using this method removes the problems associated with qualitative assessments like SCORAD, PP-NRS, and EASI, which are widely used in the clinical setting.

To address this clinical need, a SensorIsed Glove for Monitoring Atopic dermatitis (SIGMA) was developed as a low-cost, convenient, and acceptable way of assaying scratches [16] as a proxy to itch. This allows for an objective method of quantifying scratches in patients with moderate-to-severe AD at the comfort of one’s home, without requiring camera surveillance. SIGMA aims to provide subjects an alternative to the intrusiveness and discomfort by forgoing the use of cameras, instead using a combination of electronic sensors and machine learning to predict scratching. It could also be used as a complementary device to video recordings by reducing the time-consuming process of observing the video for scratch bouts.

## 2. Materials and Methods

### 2.1. SensorIsed Glove for Monitoring Atopic Dermatitis (SIGMA)

SIGMA (Figure 1) is fabricated using an off-the-shelf biking glove of 92% polyester, 8% elastane. Microtubular stretch sensors [17] were mounted onto the superior side of the glove over each finger with Velcro for ease of repositioning them for different subjects. The whole device, including its control unit, is considerably light, weighing 45 g in total. The customised microtubular sensors are silicone microtubes filled with Eutectic gallium indium (eGaIn) liquid metal and manufactured by Microtube Technologies Private Limited. Each microtube sensor is flexible, stretchable, robust, and washable [18], and weighs around 2 g. The 0.16 mm microchannels in the microtubular sensor contract and expand as the sensor is stretched or relaxed, respectively, causing a change in the electrical continuity of eGaIN at a rate of 3.27±0.08 MS/m, up to 220% in length [17].

### 2.2. Controller Unit

These microtube sensors were connected by wires to a 3D-printed case housing a controller unit consisting of a 370 mAh 3.7 V lithium-ion battery, a single printed circuit board integrating a microcontroller (MCU), a Bluetooth Low-Energy (BLE) transmitter, an inertial measurement unit (IMU), and a 24-bit analogue-to-digital converter (ADC). The ADC translates the change in resistance from the stretch sensors on each finger to data readable by the MCU, along with the accelerometer data (x, y, and z axes) from the integrated IMU, which were packaged and transmitted via BLE to a computer and read with a dedicated Windows 10 companion application. The companion application written in Unity3D allowed for basic visualisation of the live raw data in a chart, and saved the raw data onto the computer for further processing. The data for the stretch sensors, as well as the accelerometer data, were sampled at 20 Hz. The controller unit is 5 × 3.5 × 1 cm in dimensions and weighs in at 25 g. The casing serves to protect the circuitry and the subjects from coming into contact to mitigate the risk of an electric shock, and it is adhered onto the dorsal side of the glove using Velcro.

### 2.3. Experiment Design for Model Training

A machine learning model was developed for SIGMA to detect scratching; an experiment was designed to collect raw data from SIGMA to train this model. A total of five subjects were recruited for this study. They were healthy adults aged between 21 and 60 with no major chronic illnesses or pre-existing conditions that would prevent them from wearing gloves or understanding instructions required in the conduct of the experiment. The study was approved by National University of Singapore Institutional Review Board (NUS-IRB). The subjects donned SIGMA on their right hand, and the microtube sensors were positioned to stretch from the subject’s distal interphalangeal (DIP) joint across the dorsal side of their fingers past the metacarpophalangeal (MCP) joint.

The experiment focused on the collection of data from 100 different types of scratching and non-scratching actions for training and verifying the model, as proposed by Chun et al. [14]. Each action, scratching or non-scratching, would be performed for 10 s from resting state to resting state for the training. The non-scratching actions were to sit idly, perform different types of waving, type on a mobile phone and on a keyboard, and tap on desks in multiple different ways. Additional actions from domestic activities of daily living, such as picking up and moving objects, capping a bottle, turning a key, etc. (Figure 2A), were also included. These are common actions that a person would perform throughout the day [19], and including them in the training would allow the model to identify these actions. The scratching actions consisted of two types: arm-dominant scratching and finger-dominant scratching; additionally, each of the two types were further split into “scratching normally” and “scratching intensively”. It was thought that the change in intensity may affect how the subject would approach the action. An additional type of action, rubbing, had been added and split into “rubbing normally” and “rubbing intensively” as with finger or arm dominant scratching. Rubbing is a common replacement for scratching as it could also trigger the neurological pathways that relieve itching [20]. It was thought to be crucial that rubbing should be included in case the model was unable to distinguish between rubbing and scratching actions [13]. Each of these three types of actions would be performed on multiple areas of the body, such as the calves, the stomach, and the top of the head. By collecting rubbing, along with the two types of scratching in different intensities, the model could effectively differentiate these actions from the sensor and accelerometer data, making it more robust. The list the actions collected to train the model can be found in the Appendix A.

Finally, a separate single sequence of actions was taken for validation. The sequence is in the following order: scratching intensively, scratching normally on the back of the hand; scratching intensively, scratching normally again on the forearm, and finally, opening and closing the hand in the air rapidly then normally. This was the validation set that is used to verify the trained model, which includes an unknown action, opening and closing the hand in the air, to test the robustness of the training. The validation set was also labelled by a two-digit subject identification number, followed by the action number.

### 2.4. Data Processing for Model Training

The data were processed to extract the relevant features for training the model; the features of note were quick oscillations on one or multiple microtube sensors for finger-dominated scratching, and curled fingers and quick oscillations on the IMU accelerometer for arm-dominant scratching. The raw data—microtube sensor data of each of the fingers; acceleration data in the x, y, and z-axis from the IMU—were processed using a 0.5 Hz, 4th-order high-pass Butterworth filter (Figure 3, in blue). The intention was to remove low-frequency features, such as drift in the accelerometer, or noise due to unintentional movement in the fingers flexing or extending when the subject repositions their upper limb. Deliberate scratching movements of the fingers were observed to vary between 3–8 Hz for both intensive and normal scratching (Figure 4, also in Blue). Similar frequency ranges were observed in the accelerometer for arm-dominant scratching. A second process (Figure 3, in red) parsed the 5 microtube sensor data through a 0.03 Hz high-pass filter to remove sensor drift, followed by a moving average filter: a window of 30 samples at a 20-sample period to smooth out signals that have had rapid changes such as scratching.

Finally, the processed data were placed on a 10 mΩ threshold in magnitude post-filtering. This allows us to detect if each of the fingers were in the “bent” or “extended” state (Figure 4, also in Red). It is known that when one scratches, their fingers curl with the fingernails at an approximate angle of 80 degrees with reference to the palm [21]. Altogether, the data processing generates 13 sets of processed data using 8 sets of raw data, which will be fed into the model for training and verification.

### 2.5. Artificial Neural Network Model

A convolutional neural network (CNN) is a deep learning algorithm that can take in an input image or a matrix of data and assign importance, e.g., learnable weights and biases, to various aspects/objects in the image to be able to differentiate one from the other [22]. Long short-term memory (LSTM) is an architecture with an appropriate gradient-based learning algorithm designed to overcome backflow issues [23]. LSTM can remember the preceding states for both short times (short-term), as well as preceding states for longer times or terms [24]. A combined CNN-LSTM proved to be a highly accurate and reliable model with high accuracy [25,26] that could perform timed analysis of the features extracted in CNN. The following customised CNN-LSTM model was conceptualised and was used for training the model. The data processing, training, and verification were performed on Python 3.9.12. The CNN-LSTM model was built with the help of Keras 2.9.0 and Tensorflow.

### 2.6. Experiment Design for Pilot Clinical Study

This was a pilot single-centre study to evaluate SIGMA’s detection of scratching, compared with direct visualisation recorded using a mobile device in a group of children with moderate-to-severe AD according to the Hanifin–Rajka criteria [27]. Inclusion criteria are children aged 4 to 12 years old with an EASI score of 8 or more. The subjects donned the SIGMA glove for up to 30 min and were filmed for the duration. Children having active eczema with secondary infection or impetiginisation, other skin conditions that may also cause itch, neurodevelopmental impairment or neurobehavioural issues, or allergies to synthetic fibres were excluded from the study. Should they exhibit allergic reaction at any point of time during the study, they will immediately have SIGMA removed and treated and subsequently be dropped out. Subjects were identified by the paediatric dermatology clinic of the KK Women’s and Children’s Hospital and consent was sought with their parents or legal guardians. The subjects’ medical information such as their EASI score, PP-NRS, age of onset, treatment, and treatment responses were recorded. The subject and their caregivers were instructed on the use of SIGMA. Upon the subject donning SIGMA on their dominant hand, the subject was provided with books or videos. After 30 min, SIGMA was removed from the subject and the data and video recording were collected for post-analysis.

After each session, the electronics were removed and disinfected with 70% isopropyl alcohol, and the glove was washed in mild soap and warm water and air-dried before another subject would don the glove, in the hope of mitigating potential flare-ups. Multiple pairs of gloves were available to facilitate the trial in case it would not dry in time.

## 3. Results

### 3.1. Model Training

Data windows of 7 s for the 13 processed data matrices for each action, justified from the centre (Figure 4, processed data highlighted in grey) were extracted to account for transition to and from resting during the experiment; they were then individually sectioned into 30-sample windows at 20 samples or 1 s intervals for every action, which coincided with the sampling rate of SIGMA. Each window consisted of the sample at the point of sampling and 29 samples before it, and any window that was less than 30 samples was discarded. Every window was then labelled as either 0: non-scratching or 1: scratching for the type of action it belonged to, with all scratching and rubbing actions being 1, and 0 for all other actions. The windows from all the actions were then stacked together into an array in preparation to train the model. The customised CNN-LSTM model (Figure 3) was then trained using the array and saved for validation.

### 3.2. Post-Training Validation Test

The validation dataset, a separate data set was manually labelled with 0: non-scratching or 1: scratching at each data point by observing the data in relation to the sequence of actions. The validation include four scratching actions, and two non-scratching actions that were not part of the actions used for training: the opening and closing of hands. They were then sectioned into the same 30-sample windows at 20-sample periods and then stacked as an array, windows of fewer than 30 samples were discarded, as with the training set. Each window was labelled (0 or 1), which was derived from the median of all labels in the window. This accounts for windows in transition between non-scratching and scratching in the validation set, as it was a sequence of actions that has both scratching and non-scratching actions.

The validation data window was then fed through the trained model, which output either 0: not scratching or 1: scratching for every window (Figure 5). Comparing the model output against the label, the model attained a test accuracy of 83.6%, with sensitivity and specificity of 83 and 84%, respectively, using the validation data (Figure 6).

### 3.3. Thirty-Minute Validation Test

An additional validation test, where a healthy adult subject was asked to don SIGMA and play a smartphone game for 30 min. The subject was prompted to scratch momentarily at 4 min intervals by the conductor. This validation test sets up the future scenario that the glove was envisioned to be used in. Children with AD would don SIGMA in a controlled environment, such as a consultation room in a hospital, to quantify their scratching. Consideration was made, and it was decided that some form of interaction with smartphones [28,29] was the most likely scenario in the environment that was envisioned.

The dataset was labelled manually and was processed similarly to the post-training validation dataset (Figure 7). In this scenario, the model could predict with 99% accuracy and specificity, with a sensitivity of 74%. A single prediction error happened at around the 16 min mark (Figure 8). This validates the effectiveness of this model for detecting scratching patterns.

### 3.4. Pilot Clinical Study

A total of 8 children (8.6 ± 1.9 years old) were enrolled for the pilot study. Seven subjects completed the trial, with one subject opting to drop out 10 min into the trial. One subject’s data were incompletely recorded due to the hardware problem during the trial. A total of six subjects’ data were analysed.

The data collected from SIGMA were analysed by feeding them through the model as described in the 30 min validation; the video recordings were viewed to find exact time points where the subject scratched and cross-correlated with each other (Figure 9). Using the data to form a confusion matrix, the sensitivity, specificity, precision, negative prediction value (NPV), and accuracy could be derived by comparing the labels from the model against the time points of the accompanying video recording (Table 1).

Specificity and NPV were the highest, largely due to large numbers of true negatives, which also contributed to high accuracy value. Precision and sensitivity were lower due to large numbers of false positives caused by non-scratching movements. Two subjects, S5 and S8, had NULL for their sensitivity. This was due to them scratching exclusively with their non-dominant hand, which SIGMA was not worn on. One subject, S7, scratched with both hands and SIGMA could detected only the scratches made by the donned hand. S7 also made many actions that the model classified as scratching, contributing to the high scratching time.

## 4. Discussion

In both the post-training validation test and the 30 min validation test, the model was able to predict actions of scratching with high accuracy. We found that the post-training accuracy decreased to 83% due to the unknown action of opening and closing the hand in the air, which was included to test the robustness of the model. The model detected opening and closing the hand quickly as scratching, which was false-positive, while opening and closing slowly was correctly labelled as non-scratching (Figure 5). This phenomenon is not consistent between the subjects, with some having opening and closing their hands normally being predicted as scratching. The deduction is that performing the action quickly is analogous to finger-dominant scratching in the profile, in which the speed, or rate of change, matters between it being classified as scratching or not. This problem, however, is not seen in the 30 min validation test, as all the actions are known; therefore, the accuracy was high: 99%. Another observation was that there were discrepancies between labelling manually versus model prediction (Figure 5). Manual labelling depended on visual inspection of the data, while the model depended on the windowing function (Figure 4). It was inconclusive whether manually labelling or the model prediction were closer, as no video recordings were taken. However, both methods were largely in concordance with the unknown actions as exceptions.

The data process with the threshold (Figure 4, in red) was a retroactive addition, and was due the model predicting known hand-waving data as scratching. Examination of the data showed that the two data profiles were similar, as they both consisted of the subject moving their arm rapidly side-to-side. Okuyama et al. [21] presented absolute angles of the index finger when a single subject was scratching, and the team initially wanted to incorporate angles of all the fingers. However, due to having multiple subjects with varying finger lengths, it would be tedious to derive absolute angles of every finger for every subject. Therefore, a simple threshold of 10 mΩ or 3 mm of extension after processing was used instead to split the fingers between “bent” and “extended”. The model could then effectively separate hand waving from arm-dominant scratching, largely without issues. The 30 min validation test (Figure 7) also showed multiple instances of the subject having “bent” fingers, but it was not detected as scratching. Scratching intensively and normally on the same part of the body did not have visually observable changes in the pattern other than having higher frequency during intensive scratching. It showed that the subjects did not change their scratching method despite the difference in instruction. As the study was focussed on training a model to detect scratching, it should help in future if subjects exhibit different types of scratching or rubbing due to an intensity change.

Rubbing was also found to be similar to scratching in when presented data profile. Subjects rubbing with the tips of their fingers had profiles akin to finger-dominant scratching, while rubbing with their arms mimicked arm-dominant scratching. One benefit of including rubbing was that in some of the rubbing data collected, there were ones with high x-axis accelerometer activity, which none of the normal or high-intensity scratching exhibited. We believe this is due to the larger variation in rubbing as opposed to scratching where limited by requiring fingernails to be in contact with the skin, thus limiting the axes of movements. As rubbing exhibited a very similar profile to scratching, and performs the same role as scratching, it could therefore be considered as scratching, and has been grouped together with scratching.

The experiment to train SIGMA had no plans of having any video recordings, as it was a controlled experiment with distinct intervals where subjects performed predefined actions and rested their SIGMA-donned hand in between; therefore, labelling could be performed solely by observing the data. One noticeable downside was that through SIGMA data, it could not determine where the subjects had scratched themselves unless it was known beforehand. It was also observed that some subjects would perform scratching from using only one finger rather than using all five fingers for different parts of the body. The setup of having microtube sensors on each finger allowed for such observations. The fidelity of being able to detect each finger movement individually could have other purposes beyond the scope of this study.

The pilot study with child subjects exposes SIGMA and its ML model to the real world, where unlike the 30 min validation test, they were subjected to predicting an inexhaustive variation of actions that the subject could perform as scratching or not scratching while awake. This could be seen in low sensitivity (40.4%) and precision (6.3%), where the values were lowest. While the accuracy of all the subjects was overall high (82.8%), this could be due to the large numbers of true negatives that contributed to it, as seen by the high specificity (84%) and NPV (99.1%) values (Table 1).

However, SIGMA was able to detect the video-recorded scratches, which were true positives, to a high degree of accuracy, provided they were scratching exclusively with the hand with SIGMA donned. Omitting subjects S5, S7, and S8, where S5 and S8 had performed all their scratches with their non-dominant hand while S7 had scratched with both hands, the glove was able to detect scratching (true positives) at 94.4%. Once again, discrepancies between labelling manually versus model prediction could be attributed to a loss in concordance, but this did not have an impact in finding true positives. Although the trained ML model was observed to be able to reject a multitude of actions, it was also capturing a lot of non-scratching actions as scratching (false positives). As seen in the 30 min validation test, the unknown action of opening and closing hands, read as false-positive, was also observed here; other actions, such as pinching the book corner and flipping it, gripping the edges of the chair or table, and many more actions, were falsely captured as scratching. The model could be further trained to differentiate these actions and reject them as scratches, but it would be impractical to train for the many actions and their variations that would make the model practical to assay scratching during the daytime, where the subject is actively using their hands for all sorts of activities.

This is the probably a reason why most of the reviewed studies of devices that try to objectively measure scratching gravitate towards nocturnal studies, where voluntary use of the hands is minimised. Sleeping subjects would also be less self-conscious, as well as not preoccupied with activities that could prevent them from scratching.

Comparing scratching time with PP-NRS and EASI scores, correlation between scratching captured by video recording or SIGMA were unfounded. The short trial duration of 30 min and the preoccupation of the subjects that may also have contributed to this finding.

## 5. Conclusions

The development of SIGMA to detect scratching with a machine learning model has demonstrated good results. Its shortcomings were also apparent: that it would not supersede video recording as a gold standard in assaying scratches. SIGMA can instead be used as a complement to video recordings, especially if the experiment time is long, to help pinpoint time(s) when a subject is scratching, and researchers can advance to the time(s) and observe where the user is scratching. It can also possibly cover gaps in camera angles when the subject seems to be scratching but concise observations of scratching at the instance are not possible. SIGMA could facilitate this by assisting clinicians to capture scratches for them to use in enhancing the understanding of itch-causing conditions, as well as monitoring treatment response and drug development.

This study also gave light to the issues of performing scratch studies during the daytime with conscious subjects. While SIGMA was able to detect true scratching events that are confirmed by video to a very high degree, it was also characterising a lot of other activities and hand movements as scratching, leading to low sensitivity and precision when compared with video. Therefore, it infers that work to make non-video-recording-based scratch detection devices needs to be more elaborate in sensing and processing in order for it to be able to perform during daytime with a large variety of actions that a person may perform. The issue of conscious subjects preferring to endure the itch and not scratch is also present.

The team also considered having the ML model separate scratch prediction into intensive scratching/rubbing and normal scratching/rubbing in the future to investigate if there would be a difference in the scratching intensities for subjects of different severities in clinical assessments.

## Figures and Tables

**Figure 1 sensors-23-09782-f001:**
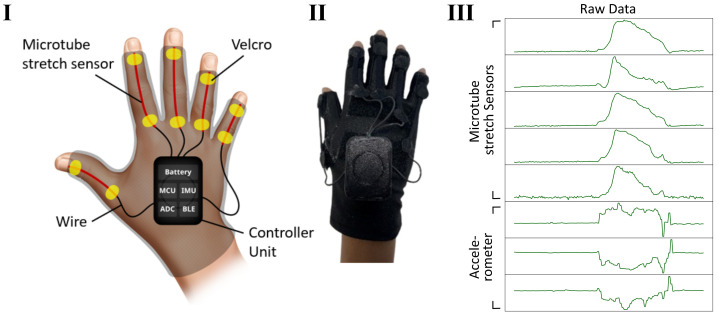
The schematic (**I**) and an image of SIGMA (**II**), and raw data (**III**) collected from SIGMA from opening and closing the hand in the air.

**Figure 2 sensors-23-09782-f002:**
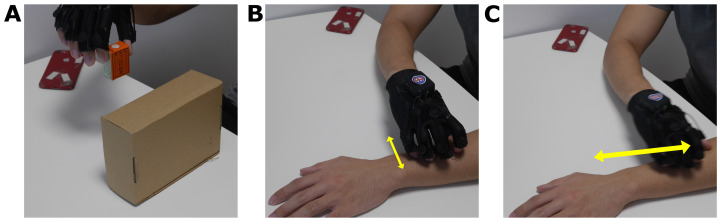
Examples of actions performed: (**A**) Moving a small object, (**B**) Finger-dominant scratching, (**C**) Arm-dominant scratching.

**Figure 3 sensors-23-09782-f003:**
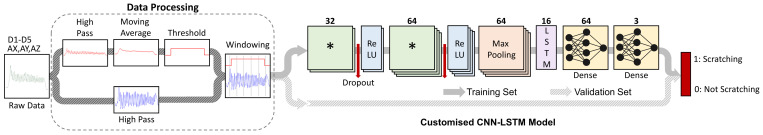
Data process flowchart. The 5 microtube sensor data and the data from the 3 axes of the accelerometer were processed with a 4th-order, 0.5 Hz high-pass Butterworth filter (data in blue). A second data process parses only the 5 microtube sensor data though a 4th-order, 0.5 Hz high-pass Butterworth filter, then a moving-average filter of 30 samples at a step of 20 samples, and finally a threshold of 10milliOhms (data in red). The customised CNN-LSTM model consists of a 32-filter convolution layer (demarcated by a *) with drop-out and ReLU, another 64-filter convolution layer with drop-out and ReLU, a 64-filter pooling layer, a 16-filter LSTM layer, a 64-filter dense layer, and finally, a 3-filter dense layer. The training data will be fed through the model to train and saved separately, and the validation data will use the saved model to predict scratching.

**Figure 4 sensors-23-09782-f004:**
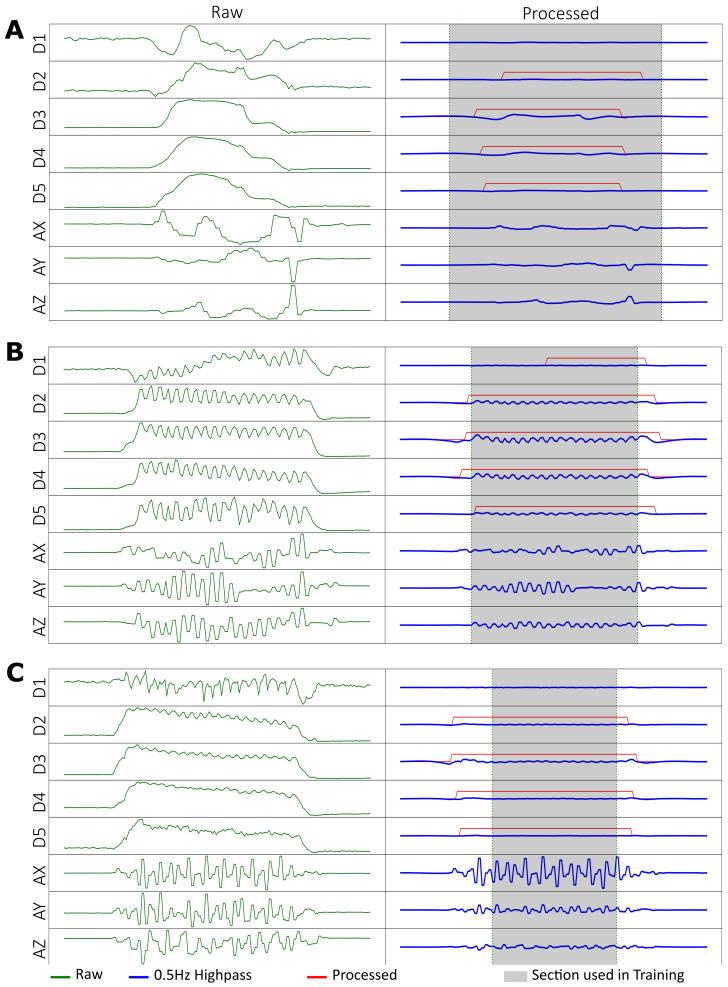
Raw and processed data of actions: (**A**) transferring a small object, (**B**) finger-dominant scratching, (**C**) arm-dominant scratching. Refer to Figure 2 for photos of the actions. Highlighted in grey in the “Processed” column are 7 s of the data window that will be used in training.

**Figure 5 sensors-23-09782-f005:**
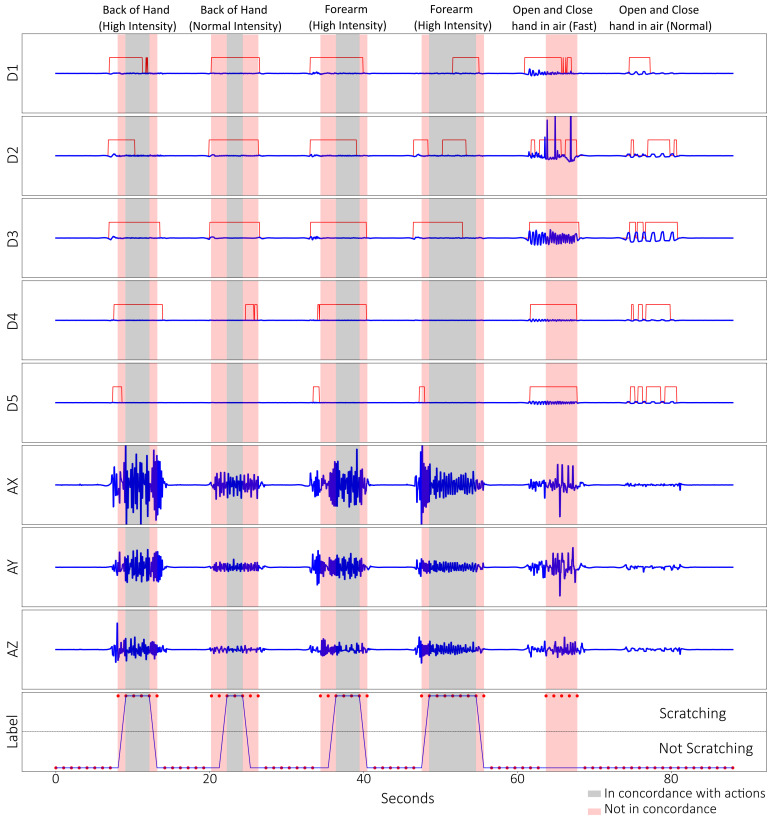
Figure of the processed data and labels of a post-training validation set from one of the subjects. The 10 processed data from the 5 microtube sensors in red and blue lines: D1 = thumb, D2 =index, D3 = middle, D4 = ring, D5 = little; the processed accelerometer data: AX = X-axis, AY = Y-axis, AZ = Z-axis; as well as the labels: red dots = prediction by ML model, blue Lines = Manually labelled. Highlighted in grey are where both ML prediction and manually labelled scratching occur, while red highlights are not in concordance.

**Figure 6 sensors-23-09782-f006:**
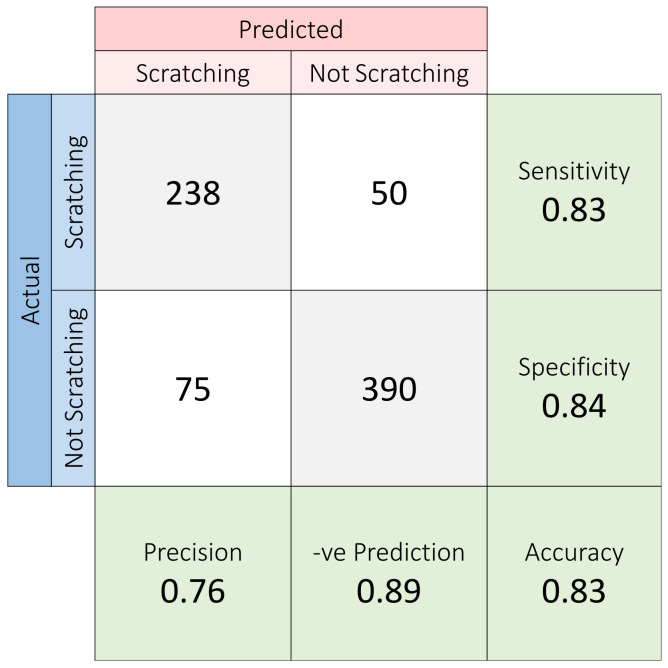
Confusion matrix of post-training validation test.

**Figure 7 sensors-23-09782-f007:**
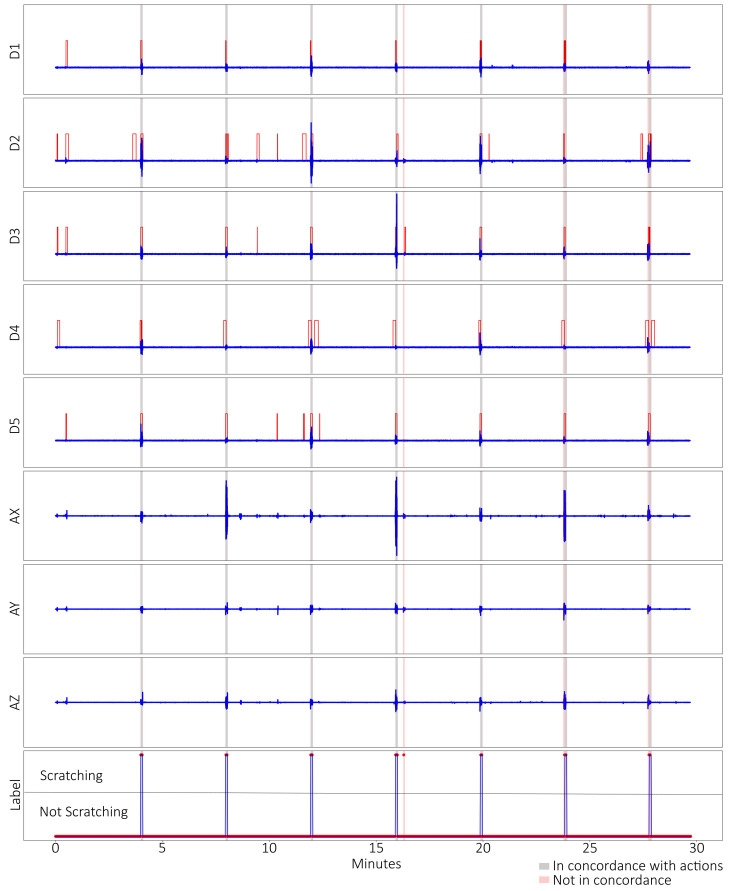
Processed data and labels for the 30 min validation test. The processed data from the 5 microtube sensors in red and blue lines: D1 = thumb, D2 = index, D3 = middle, D4 = ring, D5 = little; the processed accelerometer data: AX = X-axis, AY = Y-axis, AZ = Z-axis; as well as the labels: red dots = prediction by ML model, blue Lines = manually labelled. Highlighted in grey are periods where both ML prediction and manually labelled scratching occurs, while red highlights are not in concordance.

**Figure 8 sensors-23-09782-f008:**
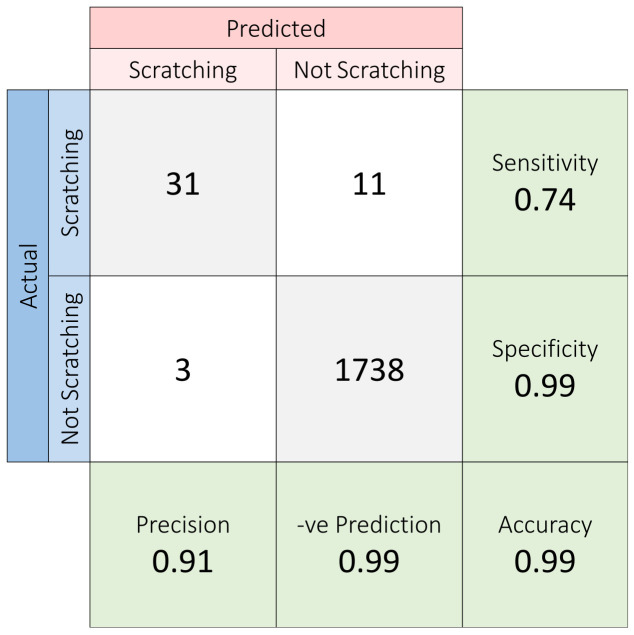
Confusion matrix of 30 min validation test.

**Figure 9 sensors-23-09782-f009:**
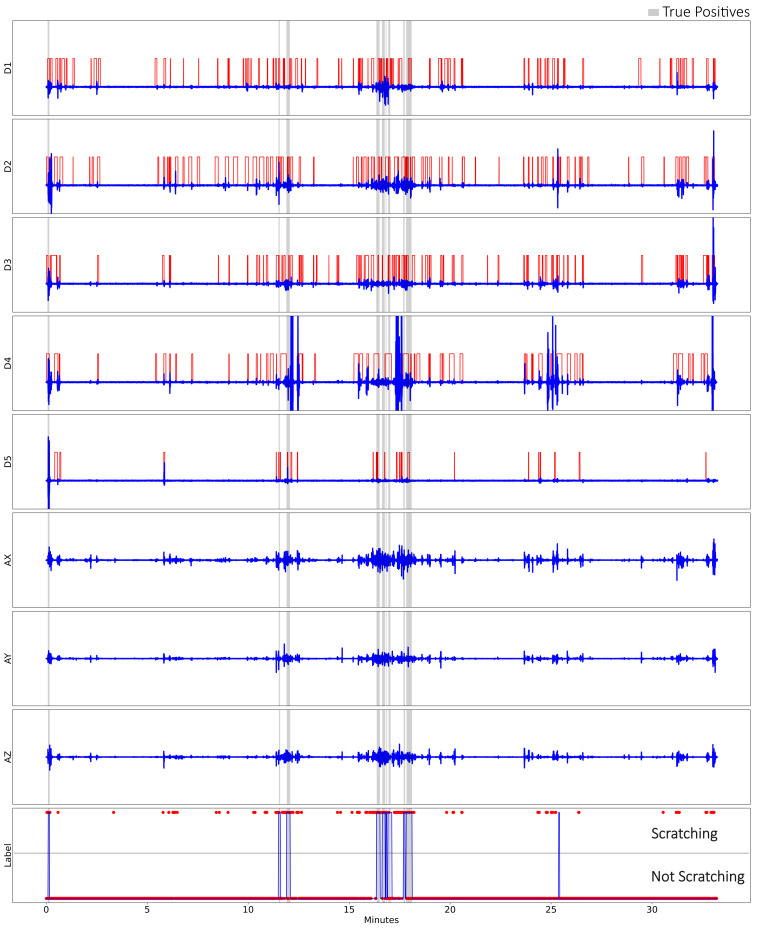
Processed data and labels for Subject S4 in the pilot clinical study. The processed data from the 5 microtube sensors in red and blue lines: D1 = thumb, D2 = index, D3 = middle, D4 = ring, D5 = little; the processed accelerometer data: AX = X-axis, AY = Y-axis, AZ = Z-axis; as well as the labels: red dots = prediction by ML model, blue lines = manually labelled.

**Table 1 sensors-23-09782-t001:** Parameters from the subjects in pilot clinical study.

	S3	S4	S5	S6	S7	S8	Mean
PP-NRS [0–10]	2	5	8	3	5	5	5 ± 2
EASI	8	16.4	23.5	8.2	8.2	14.4	13.1 ± 6.2
Total Scratch Time (Video) [s]	77	72	92	31	20	20	52 ± 32
Total Scratch Time (Video) [%]	0.04	0.04	0.05	0.02	0.01	0.01	0.03 ± 0.02
Total Scratch Time (SIGMA) [s]	262	186	184 ${}^{★}$	270	815 ${}^{\wedge }$	104 ${}^{★}$	303 ± 258
Total Scratch Time (SIGMA) [%]	0.14	0.9	0.1	0.15	0.45	0.05	0.16 ± 0.14
Sensitivity	0.21	0.52	NA	0.41	0.47	NA	0.4 ± 0.14
Specificity	0.86	0.92	0.9	0.86	0.55	0.94	0.84 ± 0.14
Accuracy	0.82	0.91	0.9	0.85	0.55	0.94	0.83 ± 0.14
Precision	0.09	0.23	0	0.06	0.01	0	0.06 ± 0.08
Negative Predictive Value (NPV)	0.98	0.98	1	0.99	1	1	0.99 ± 0.01
%True Positives Detected (SIGMA)	0.89	1	0 ${}^{★}$	0.94	0.5 ${}^{\wedge }$	0 ${}^{★}$	0.56 ± 0.46

${}^{★}$ Values consist only of false positives as subject only scratches with the hand without SIGMA. ${}^{\wedge }$ Values are mostly false-positive as the subject was scratching with both hands.

## Data Availability

The data presented in this study are available on request from the corresponding author. The data are not publicly available due to the video recording that could compromise the privacy of research participants.

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
