# Peer review of "A Sensorised Glove to Detect Scratching for Patients with Atopic Dermatitis"

_sensors, 2023, doi:10.3390/s23249782_

Round 1
Reviewer 1 Report
Comments and Suggestions for Authors
Summary of findings: The authors developed a sensor-attached glove system to detect scratching that occurs due to Atopic Dermatitis. They addressed that their system is effective in detecting Atopic Dermatitis by using a modified CNN-LSTM-based machine learning algorithm. They used a developed microtubular sensor system to collect data for classification. They did not propose any new sensing systems, additional experiments, or any innovative design approach; therefore, I believe this study still lacks novelty. They also analyzed, in some cases, where their systems made incorrect classifications, which could reduce real-world applicability. Therefore, I ask the authors to address the following comments for publication at MDPI.
detailed comments:
1. In the introduction section, in line 2, they refer to periods where symptoms of Atopic Dermatitis are more severe. They can provide more details about these periods so that readers can gain a better understanding of Atopic Dermatitis.
2. In line 23, they provide some methods currently used to assess the severity of Atopic Dermatitis but do not include any references for those methods.
3. The format of some references incorrect (e.g., ref. 3-4), and some references are old (e.g., ref. 5). They need to provide more current references.
4. In the introduction section, please provide working principles of SIGMA and how it can be an alternative of clinical assessments. Are there any other works for quantitative assessments of scratching?
5. Please provide the characteristics of the microtubular stretch sensors such as stretchability and resistance changes over stretching in the Materials and Methods section.
6. Discuss features used for the classification.
7. Based on Figure 3 and 6, it looks scratching (e.g., finger and arm-dominant) can be detectable only with IMU. Have you compared the classification accuracies using IMU+microtube sensor data and only IMU data?
8. In line 139, they addressed their customized CNN-LSTM model. Can they prove that their customized model performs better in the classification by comparing it with the existing CNN-LSTM or RNN-LSTM model mentioned in line 137?
9. In line 182, they stated 83% accuracy with 83% sensitivity. Can you discuss possible solutions to increase the sensitivity?
10. It looks data imbalanced for 30-minutes validation test, that may be why the accuracy and sensitivity are different from post-training validation test. Can you address the imbalance and check if the accuracy and sensitivity are changed?
11. In line 223, they mentioned that their device detects opening hand and closing hand as scratching. This questions the applicability of the model in the real world because it will produce more false positives. Is it possible to address this issue and solve it to produce more accurate sensitivity?
12. In line 262, the authors mentioned about scratching with one finger. Can you show how the microtube sensors help to improve the detection?
13. In line 285, the authors claimed it would be impractical to use the glove to detect scratching during life activities. Then what are the significance and novelty of this work?
Comments on the Quality of English Language1. Moderate proofreading/grammer correction are necessary.
2. In general, sentences are too long to understand.
Reviewer 2 Report
Comments and Suggestions for Authors
In this work, Sensorised Glove for Monitoring Atopic Dermatitis (SIGMA), a lightweight compliant glove that detects scratching using data from microtubular stretchable sensors, an inertial measurement unit (IMU) and a machine learning model is presented. The machine learning model was trained with experimental data. The work is interesting. However, following points should be addressed before publication:
1) How the microtubular stretch sensors are fabricated. Are they commercially available or fabricated by the author in lab? The details of sensor development is missing in manuscript and should be included for the readers.
2) What is the size of microchannels in microtubular stretch sensors.
3) How does the dimensions of microtubular stretch sensors affect the sensitivity and other sensing parameters.
4) The inertial measurement unit (IMU) measures the change in resistance. The details of circuit diagram along with signal processing unit should be included in manuscript or Supplementary file.
5) Authors conducted a pilot single centre study to evaluate SIGMA’s detection of scratching in a group of children with moderate to severe AD. Ethical permission number if so for performing the study can be mentioned.
6) What are the low and high risk associated with this?
Comments on the Quality of English LanguageManuscript should be rechecked for grammatical errors.
